# MaPPO: Maximum a Posteriori Preference Optimization with Prior Knowledge

## Abstract

As the era of large language models (LLMs) on behalf of users unfolds, Preference Optimization (PO) methods have become a central approach to aligning LLMs with human preferences and improving performance. We propose Maximum a Posteriori Preference Optimization (MaPPO), a framework for learning from preferences that explicitly incorporates prior reward knowledge into the optimization objective. While existing methods such as Direct Preference Optimization (DPO) and its variants treat preference learning as a Maximum Likelihood Estimation (MLE) problem, MaPPO extends this paradigm by integrating prior reward estimates into a principled Maximum a Posteriori (MaP) objective. This not only generalizes DPO and its variants, but also enhances alignment by mitigating the oversimplified binary classification of responses. More importantly, MaPPO introduces no additional hyperparameter, and supports preference optimization in both offline and online settings. In addition, MaPPO can be used as a plugin with consistent improvement on DPO variants, including widely used SimPO, IPO, and CPO. Extensive empirical evaluations of different model sizes and model series on three standard benchmarks, including MT-Bench, AlpacaEval 2.0, and Arena-Hard, demonstrate consistent improvements in alignment performance without sacrificing computational efficiency.

## 1 Introduction

Reinforcement Learning from Human Feedback (RLHF) has emerged as a general paradigm for aligning large language models (LLMs) with human preferences. Pioneering work framed the problem as reinforcement learning (RL) on a reward model trained from group-wise comparisons, yielding notable improvements in summarization and dialogue (Christiano et al., 2017; Stiennon et al., 2020). Subsequent systems such as InstructGPT (Ouyang et al., 2022) demonstrated that RLHF could scale to billion-parameter models and substantially boost helpfulness and safety (Lan et al., 2025). Despite its practical successes, RLHF still suffers from noisy feedback, reward-model misalignment, optimization instability, and computation inefficiency with high memory cost, which together hinder its scalability and reliability (Casper et al., 2023; Dai et al., 2023; Pan et al., 2022).

Direct Preference Optimization (DPO) (Rafailov et al., 2023) reframes the preference learning with a Kullback–Leibler (KL) regularized objective as a log-odds objective, effectively turning the task into Maximum Likelihood Estimation (MLE) over pairwise comparisons: The model is trained to assign a higher likelihood to the preferred response than to the rejected, while staying close to a reference policy. The MLE perspective accounts for the efficiency of DPO, as it eliminates the need for rollouts or value functions. However, this formulation also introduces a fundamental limitation: It considers only the relative likelihoods within each pair, overlooking the absolute reward magnitude and any external prior knowledge (Amini et al., 2024; D'Oosterlinck et al., 2025). As a result, the training signal in DPO is inherently local, bounded by pairwise comparisons, and lacks global calibration across examples.

**Challenge.** A fundamental limitation of MLE-based preference optimization lies in its purely relative nature: It focuses on maximizing the *gap* between chosen and rejected responses, yet lacks a mechanism to anchor their absolute probabilities. As training progresses, the MLE objective tends to suppress the likelihood

of the rejected response rather than elevate that of the preferred one. Empirical investigations (Pal et al., 2024; Rafailov et al., 2024; Tajwar et al., 2024; Zhang et al., 2024) consistently show a simultaneous reduction in the absolute probabilities assigned to both preferred and rejected answers, resulting in abnormal output distributions. This undesirable dynamic, empirically known as the *squeezing effect* (Ren & Sutherland, 2024), undermines policy calibration and injects instability into generation. The issue is especially severe in near-tie cases (in Figure 2), especially when models approach human-level performance (Liu et al., 2024a; Guo et al., 2024), where both responses are reasonable yet MLE still enforces an artificial separation, draining probability mass from the high-quality region of the output space. Motivated by this, the key question that this paper aims to address is:

*How can we improve language model alignment through a more principled training signal, instead of an oversimplified MLE pipeline?*

In this paper, we answer the above question by introducing Maximum-a-Posteriori Preference Optimization (MaPPO), a simple yet principled extension of DPO that injects data-driven prior knowledge into preference training. MaPPO augments the standard maximum-likelihood objective with a lightweight MaP regularizer, an additional log-prior scaled by a calibrated reward gap, which proportionally adjusts each update to the confidence difference between the preferred and rejected answers. Instead of the oversimplified binary classification in MLE, this mechanism curbs the excessive penalization of near-tie pairs while preserving DPO's one-step closed form and computational efficiency.

Extensive experiments demonstrate that MaPPO delivers consistently stronger performance across three public alignment benchmarks: AlpacaEval 2.0, Arena-Hard, and MT-Bench. We evaluate MaPPO on multiple model families, including Llama-3, Qwen2.5, and Mistral, under multiple model sizes. Compared to DPO, MaPPO achieves absolute win-rate gains of 94.3% on AlpacaEval and 37.1% on Arena-Hard when fine-tuned on the Mistral-7B-Instruct model. Moreover, the proposed MaPPO is suitable for both offline and online settings. These results validate that a lightweight prior is sufficient to produce stronger and better-calibrated policies. Furthermore, MaPPO is designed as a drop-in regularization module and seamlessly integrates with a broad spectrum of recent DPO variants, including Iterative-DPO (Dong et al., 2024), SimPO (Meng et al., 2024), IPO (Gheshlaghi Azar et al., 2024), and CPO (Xu et al., 2024). In all cases, we observe consistent gains up to 31.3% on Arena-Hard in alignment scores without requiring additional computation or changes to the optimization pipeline. This suggests that MaPPO serves as a robust and general enhancement strategy for advanced preference training pipelines.

**Contributions.**    In summary, the main contributions of this work are as follows:

1. We propose MaPPO, a principled extension of Direct Preference Optimization, which incorporates data-driven prior reward estimates into a Maximum-a-Posteriori (MaP) objective.

2. We demonstrate that MaPPO naturally supports both offline (e.g., DPO) and online (e.g., I-DPO) preference optimization.

3. We show that MaPPO is compatible with and enhances existing DPO variants, including SimPO, IPO, and CPO. For all variants, no additional hyperparameter is needed.

4. Empirical results across multiple model series and model sizes confirm consistent improvements in alignment performance on standard benchmarks, including MT-Bench, AlpacaEval 2.0, and Arena-Hard.

**Notations.**    We use $\sigma(\cdot)$ to denote the logistic (sigmoid) function $\sigma(x) = \frac{1}{1+e^{-x}}$. For preference pairs, $\mathbf{y}_w$ denotes the *chosen* (winning) response, while $\mathbf{y}_l$ denotes the *rejected* (losing) response.

## 2    Related Work

**Direct Preference Optimization and its Variants.**    Driven by the complexity of online RLHF algorithms (Santacroce et al., 2023; Zheng et al., 2023b), recent research has pivoted toward efficient offline preference optimization. Direct Preference Optimization (DPO) (Rafailov et al., 2023) frames preference

alignment as maximum-likelihood estimation (MLE) under the Bradley-Terry (BT) model (Bradley & Terry, 1952), while IPO (Gheshlaghi Azar et al., 2024) generalizes this framework without the pointwise-reward assumption. Further, CPO (Xu et al., 2024) jointly optimizes the sequence likelihood and a contrastive reward to perform supervised fine-tuning (SFT) and alignment in one pass. KTO (Ethayarajh et al., 2024) extends the paradigm to single-response feedback via prospect theoretic utility. Recent DPO variants, ORPO (Hong et al., 2024), R-DPO (Park et al., 2024), and SimPO (Meng et al., 2024), further push performance by discarding the reference model or regularizing response length. Yang et al. (2025) further introduces a weight hyperparameter to balance the influence of preference pairs from different policies. Zhao et al. (2025) then aims to combine the previous PO methods into one cohesive objective. However, all DPO-style variants rely on MLE in the training process, which oversimplifies the tuning of preferred and unpreferred responses as a binary classification problem.

**Confidence Degeneration in DPO.** Pal et al. (2024) and Tajwar et al. (2024) show analytically and empirically that the expected DPO gradient often decreases the log-likelihood of the preferred response $\mathbf{y}_w$ instead of increasing it, leading to a simultaneous shrinkage of both responses. Rafailov et al. (2024) observe the same trend, attributing the drop to the expected log ratio between the optimized and reference models. By showing that this is equivalent to the non-negative KL divergence, they conclude that DPO training inevitably lowers the likelihood of the chosen response. Nemotron-4 (Adler et al., 2024) constraints the DPO loss with the reward margin. However, no motivation or theoretical perspective is given. Moreover, it cannot be used in the online setting and is unable to enhance the other DPO variants. More recent analyses of the learning dynamics in Ren & Sutherland (2024) have identified a phenomenon termed the *squeezing effect*, whereby DPO training aggressively drains probability mass from all responses except the most confident one, $\mathbf{y}^* = \arg\max_{i \in [V] \setminus \mathbf{y}_l} \pi_\theta(\mathbf{y} = i)$, consequently funneling this mass towards $\mathbf{y}^*$. Our method utilizes prior knowledge to soften the downward pressure on the rejected response $\mathbf{y}_l$, it markedly mitigates the squeezing effect.

# 3 Preliminary & Problem Setup

## 3.1 RL Tuning

First, we introduce the general framework of Reinforcement Learning (RL). Consider the Markov decision process (MDP) as a tuple $(\mathcal{S}, \mathcal{A}, \mathcal{P}, \mathcal{R})$, where $\mathcal{S}$ is the state space, $\mathcal{A}$ is a finite action space, $\mathcal{P} : \mathcal{S} \times \mathcal{A} \times \mathcal{S} \to \mathbb{R}$ is a Markov kernel that determines transition probabilities, and $\mathcal{R} : \mathcal{S} \times \mathcal{A} \to \mathbb{R}$ is a reward function. At each time step $t$, the agent executes an action $\mathbf{y}_t \in \mathcal{A}$ from the current state $\mathbf{s}_t \in \mathcal{S}$, following a stochastic policy $\pi$, *i.e.*, $\mathbf{y}_t \sim \pi(\cdot|\mathbf{s}_t)$. The corresponding reward is defined as $r_t$.

Following the conventional setting in LLMs, the policy $\pi_\theta$ represents the LLM with model parameters $\theta$. The action space $\mathcal{A}$ is set as the vocabulary. At step $t$, $\mathbf{s}_t = (\mathbf{x}, \mathbf{y}_{<t})$ is a cascade of the query $\mathbf{x}$ and the tokens $\mathbf{y}_{<t} = (\mathbf{y}_1, \cdots, \mathbf{y}_{t-1})$ that have been predicted, and $\mathbf{y}_t$ is the next token to be predicted. The transition kernel $\mathcal{P}$ is deterministic as $\mathbf{s}_{t+1} = (\mathbf{s}_t, \mathbf{y}_t)$. The complete answer $\mathbf{y} = (\mathbf{y}_1, \cdots, \mathbf{y}_T)$ with length $|\mathbf{y}| = T$. The step reward $r_t = r(\mathbf{x}, \mathbf{y}_{\leq t})$ can be obtained from a trained reward model.

After formalizing the LLM tuning as an RL problem, the goal of RL tuning (Ouyang et al., 2022) is to maximize the expectation of the cumulative reward $r := r(\mathbf{x}, \mathbf{y})$ with a Kullback–Leibler (KL) constraint as follows

$$\max_{\pi_\theta} \mathbb{E}_{\mathbf{x} \sim \mathcal{D}, \mathbf{y} \sim \pi_\theta(\cdot|\mathbf{x})} \left[ r(\mathbf{x}, \mathbf{y}) \right] - \beta \mathbb{D}_{\mathrm{KL}} \left[ \pi_\theta(\cdot|\mathbf{x}) || \pi_{\mathrm{ref}}(\cdot|\mathbf{x}) \right], \tag{1}$$

where $\mathbb{D}_{\mathrm{KL}}(\cdot||\cdot)$ denotes the KL divergence, and $\beta$ is a constant weight. $\pi_{\mathrm{ref}}$ is a reference model, which is usually the initial policy model before tuning. This optimization problem can be solved by any RL algorithms, *e.g.*, PPO (Schulman et al., 2017) and GRPO (Shao et al., 2024).

### 3.2 Direct Preference Optimization

In Direct Preference Optimization (DPO) (Rafailov et al., 2023), a closed-form expression of equation 1 is used, and a connection between policy $\pi$ and reward function $r$ is built as

$$\pi(\mathbf{y}|\mathbf{x}) \;=\; \frac{1}{Z(\mathbf{x})}\pi_{\text{ref}}(\mathbf{y}|\mathbf{x})\exp\big(\frac{1}{\beta}r(\mathbf{y},\mathbf{x})\big), \tag{2}$$

where $Z(\mathbf{x})$ is a partition function to normalize the probability.

With a prompt $\mathbf{x}$, we sample two responses from the current policy model $\mathbf{y}_1, \mathbf{y}_2 \sim \pi_\theta(\cdot|\mathbf{x})$. A human expert then demonstrates the preference and ranks the responses as $\mathbf{y}_w$ (win) and $\mathbf{y}_l$ (lose). After plugging in equation 2 into the reward model training (MLE) loss function, the target of RL tuning becomes to minimize the loss function shown below

$$\mathcal{L}(\theta) \;=\; \mathbb{E}_{(\mathbf{y}_w,\mathbf{y}_l,\mathbf{x})\sim\mathcal{D}}\Big[-\log\sigma\Big(\beta\log\frac{\pi_\theta(\mathbf{y}_w|\mathbf{x})}{\pi_{\text{ref}}(\mathbf{y}_w|\mathbf{x})} - \beta\log\frac{\pi_\theta(\mathbf{y}_l|\mathbf{x})}{\pi_{\text{ref}}(\mathbf{y}_l|\mathbf{x})}\Big)\Big], \tag{3}$$

where $\sigma(\cdot)$ is the logistic function. The training process can be done using mini-batch gradient descent and the variants, e.g., AdamW (Loshchilov & Hutter, 2019). Notably, the derivation on reward model training is based on MLE, which oversimplifies the process as a binary classification problem. As a result, minimizing equation 3 is nothing but increasing the gap between the output probability of $\mathbf{y}_w$ and $\mathbf{y}_l$.

### 3.3 Current issues with MLE

The learning process is to estimate the parameters $\theta$ via maximum likelihood, which is to maximize the gap of the output probability between the winning response $\pi_\theta(\mathbf{y}_w|\mathbf{x})$ and the losing response $\pi_\theta(\mathbf{y}_l|\mathbf{x})$.

Despite its computational efficiency and simplicity, the MLE formulation introduces a critical limitation: It focuses solely on relative probabilities within each preference pair, while ignoring the absolute magnitude of confidence in the responses. Shown as Figure 2, training encourages simultaneous downscaling of both $\pi_\theta(\mathbf{y}_w|\mathbf{x})$ and $\pi_\theta(\mathbf{y}_l|\mathbf{x})$, to enlarge the preference gap. This undesired tendency can lead to over-penalization of both responses, especially in near-tie cases, thereby reducing output confidence and harming policy calibration (Ren & Sutherland, 2024).

Such issues become particularly problematic as models approach human-level performance, where both the winning and losing responses may be of high quality. For example, consider a preference pair where both $\mathbf{y}_w$ and $\mathbf{y}_l$ are grammatically correct, contextually relevant, and factually accurate, with the only preference driven by stylistic nuances. In this case, the MLE objective may still enforce a large probability gap by downscaling both log-likelihoods. This inadvertently shifts the probability mass away from the high-quality response space, introducing unnecessary uncertainty into the policy, and can lead to erratic or unstable generation in downstream usage.

Table 1: Comparison of log-probabilities for responses $\mathbf{y}_w$ and $\mathbf{y}_l$ in PO methods.

| Model | logp($\mathbf{y}_w$) | logp($\mathbf{y}_l$) | logp($\mathbf{y}_w$) - logp($\mathbf{y}_l$) |
|-------|------|------|------|
| IT | $-14.3$ | $-43.4$ | $+29.1$ |
| DPO | $-121.5$ | $-443.2$ | $+321.7$ |
| MaPPO | $-10.4$ | $-321.1$ | $+310.7$ |

In Figure 1, we show an example of the prompt and response pair $(\mathbf{x}, \mathbf{y}_w, \mathbf{y}_l)$. Given the prompt $\mathbf{x}$, both $\mathbf{y}_w$ and $\mathbf{y}_l$ have correct answers, while in slightly different expressions. Both $\mathbf{y}_w$ and $\mathbf{y}_l$ have high rewards, which reflect the high qualities. However, in MLE and its derived DPO, the learning objective is nothing but to increase the gap between $\mathbf{y}_w$ and $\mathbf{y}_l$, regardless of the fact that both of them have high qualities with correct answers, and their qualities match each other. We also list an example with long responses in Appendix B. As shown in Table 1, after DPO training, the log output probability of $\mathbf{y}_w$ decreases from $-14.3$ to $-121.5$,

---

**Prompt x**

A printer prints 17 pages per minute. How many minutes will it take to print 200 pages? Express your answer to the nearest whole number.

---

**Response $\mathbf{y}_w$, $r = 0.95$**

To find out how many minutes it will take to print 200 pages, we need to divide the total number of pages by the number of pages the printer can print per minute. So we divide 200 by 17: $\frac{200}{17} \approx 11.76$. But since we want the nearest whole number, we need to round it up. That means it will take $\boxed{12}$ minutes to print 200 pages. The answer is: $\boxed{12}$

---

**Response $\mathbf{y}_l$, $r = 0.91$**

To find out how many minutes it will take to print 200 pages at a rate of 17 pages per minute, divide: $\frac{200}{17} \approx 11.76$. Rounding 11.76 to the nearest whole number gives: 12. So it will take $\boxed{12}$ minutes to print 200 pages. The answer is: $\boxed{12}$

---

Figure 1: An example of $(\mathbf{x}, \mathbf{y}_w, \mathbf{y}_l)$ pair. Both responses $\mathbf{y}_w$ and $\mathbf{y}_l$ have good quality as they achieve high rewards, where $r(\mathbf{x}, \mathbf{y}_w) = 0.95$, $r(\mathbf{x}, \mathbf{y}_l) = 0.91$, and $r \in [0, 1]$.

and $\mathbf{y}_l$ decreases from $-43.4$ to $-443.2$. Although the gap indeed increases from $29.1$ to $321.7$, it violates the principal goal: Increase the output probability of the high-quality responses.

These issues highlight the need for a more principled formulation that preserves relative preferences while incorporating global calibration and prior reward knowledge. In the next section, we introduce our Maximum-a-Posteriori (MaP) framework that addresses these shortcomings in a unified and efficient manner.

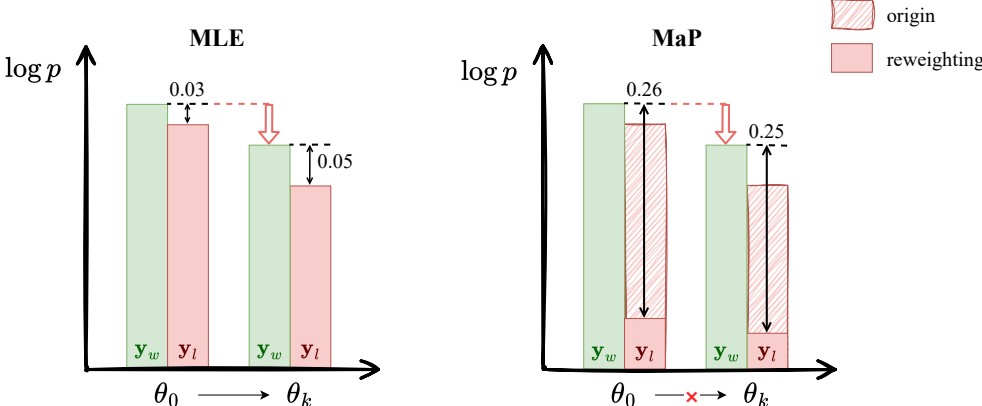

Figure 2: Under the standard MLE-based DPO (left), empirical studies (Pal et al., 2024; Rafailov et al., 2024; Tajwar et al., 2024; Ren & Sutherland, 2024) demonstrated that training tends to simultaneously downscale (with different magnitudes) both the chosen and rejected responses to increase their gap. Our MaP-based method (right) mitigates this harmful tendency by re-weighting the rejected response based on prior knowledge. Here, the x-axis denotes the initial model $\theta_0$ and a potentially harmful model $\theta_k$ that may arise during training, while the y-axis shows the log-likelihood of a fixed preference pair under different policies.

## 4 MaPPO Design

### 4.1 MaPPO Loss

In this subsection, we start the derivation step by step from the first principle.

With a prompt $\mathbf{x}$ and responses $(\mathbf{y}_1, \mathbf{y}_2)$, an oracle gives its preference on the responses as $(\mathbf{y}_w, \mathbf{y}_l)$. The Bradley-Terry (BT) model (Bradley & Terry, 1952) builds the connection between the rewards and the preference probability as follows:

$$p(\mathbf{y}_w \succ \mathbf{y}_l | \mathbf{x}) \;=\; \frac{\exp(r(\mathbf{y}_w, \mathbf{x}))}{\exp(r(\mathbf{y}_w, \mathbf{x})) + \exp(r(\mathbf{y}_l, \mathbf{x}))} \;=\; \frac{1}{1 + \exp(r(\mathbf{y}_l, \mathbf{x}) - r(\mathbf{y}_w, \mathbf{x}))}. \tag{4}$$

The preference dataset have $N$ samples denoted as $\mathcal{D} = \{\mathbf{y}_w^i, \mathbf{y}_l^i, \mathbf{x}^i\}_{i=1}^N$. We can parametrize a reward model with model parameters $\phi$ as $r_\phi(\mathbf{y}, \mathbf{x})$. Given $\mathbf{x}$, assume we have prior knowledge of rewards as $r_w$ and $r_l \in [-1, 0]$. This can be obtained from an oracle, *e.g.*, a pre-trained reward model. To incorporate the prior knowledge of rewards, we need to use the gap $\Delta_r = r_w - r_l$ as suggested in equation 4. To keep the softmax form in the BT model, we can construct the prior probability as follows

$$p(r_\phi) \;=\; \frac{\exp(r_\phi(\mathbf{y}_w, \mathbf{x})) + \exp(\Delta_r r_\phi(\mathbf{y}_l, \mathbf{x}))}{\exp(r_\phi(\mathbf{y}_w, \mathbf{x})) + \exp(r_\phi(\mathbf{y}_l, \mathbf{x}))}. \tag{5}$$

We use the reward gap $\Delta_r$ on the softmax probability to make the probability always greater than 0 and smaller than 1. Notably, this form is not unique, and other forms are also acceptable if they satisfy the properties of the probability function. We further discuss the prior function in Appendix D.1.

The MaP loss is the combination of the MLE loss and the prior knowledge loss as follows

$$\begin{aligned}
\mathcal{L}_{\mathrm{MaP}}(r_\phi) &= \mathcal{L}_{\mathrm{MLE}}(r_\phi) + \mathcal{L}_{\mathrm{p}}(r_\phi) \\
&= \mathop{\mathbb{E}}_{(\mathbf{y}_w, \mathbf{y}_l, \mathbf{x}) \sim \mathcal{D}} \Big[ -\log \sigma\Big(r_\phi(\mathbf{y}_w, \mathbf{x}) - r_\phi(\mathbf{y}_l, \mathbf{x})\Big) - \log p(r_\phi) \Big] \\
&= \mathop{\mathbb{E}}_{(\mathbf{y}_w, \mathbf{y}_l, \mathbf{x}) \sim \mathcal{D}} \Big[ -\log \sigma\Big(r_\phi(\mathbf{y}_w, \mathbf{x}) - \Delta_r r_\phi(\mathbf{y}_l, \mathbf{x})\Big) \Big].
\end{aligned} \tag{6}$$

As proved in previous works (Rafailov et al., 2023; Go et al., 2023; Korbak et al., 2022), given a reward function $r(\mathbf{y}, \mathbf{x})$, we have a closed-form expression of the policy $\pi$ as

$$\pi(\mathbf{y}|\mathbf{x}) \;=\; \frac{1}{Z(\mathbf{x})} \pi_{\mathrm{ref}}(\mathbf{y}|\mathbf{x}) \exp\big(\tfrac{1}{\beta} r(\mathbf{y}, \mathbf{x})\big), \tag{7}$$

where $Z(\mathbf{x})$ is a partition function. With a parametrized policy $\pi_\theta$, we can plug this result into the loss function equation 6, and get the MaPPO loss

$$\mathcal{L}_{\mathrm{MaP}}(\theta) \;=\; \mathop{\mathbb{E}}_{(\mathbf{y}_w, \mathbf{y}_l, \mathbf{x}) \sim \mathcal{D}} \Big[ -\log \sigma\Big(\beta \log \frac{\pi_\theta(\mathbf{y}_w|\mathbf{x})}{\pi_{\mathrm{ref}}(\mathbf{y}_w|\mathbf{x})} - \Delta_r \beta \log \frac{\pi_\theta(\mathbf{y}_l|\mathbf{x})}{\pi_{\mathrm{ref}}(\mathbf{y}_l|\mathbf{x})}\Big) \Big]. \tag{8}$$

With the MaP estimation, we achieve a clean result compared to the MLE estimation in DPO with a calibration term $\Delta_r \in [0, 1]$ from the prior knowledge.

**Remark.** In our MaPPO method, **no** additional hyperparameter is introduced compared to the original DPO method. Thus, MaPPO offers a clean and easily pluggable solution, and no extra hyperparameter tuning is needed.

### 4.2 Analysis of MaPPO

In this subsection, we analyze the connection with MaPPO, DPO, and SFT.

**Connection with SFT.**   First, in the SFT loss function, the optimal solution is equal to that from the loss function below

$$\mathcal{L}(\theta) \;=\; \underset{(\mathbf{y}_w, \mathbf{y}_l, \mathbf{x}) \sim \mathcal{D}}{\mathbb{E}} \Big[ -\log \frac{\pi_\theta(\mathbf{y}_w|\mathbf{x})}{\pi_{\text{ref}}(\mathbf{y}_w|\mathbf{x})} \Big], \tag{9}$$

where $\mathbf{y}_w$ is the supervised target.

The gradient is

$$\underset{(\mathbf{y}_w, \mathbf{y}_l, \mathbf{x}) \sim \mathcal{D}}{\mathbb{E}} \Big[ -\nabla \log \frac{\pi_\theta(\mathbf{y}_w|\mathbf{x})}{\pi_{\text{ref}}(\mathbf{y}_w|\mathbf{x})} \Big] \;=\; \underset{(\mathbf{y}_w, \mathbf{y}_l, \mathbf{x}) \sim \mathcal{D}}{\mathbb{E}} \Big[ -\nabla \log \pi_\theta(\mathbf{y}_w|\mathbf{x}) \Big] \;=\; \nabla \mathcal{L}_{\text{SFT}}(\theta). \tag{10}$$

In equation 8, when $r_w = r_l$ (*i.e.*, $\Delta_r = 0$), the loss function becomes

$$\mathcal{L}(\theta) \;=\; \underset{(\mathbf{y}_w, \mathbf{y}_l, \mathbf{x}) \sim \mathcal{D}}{\mathbb{E}} \Big[ -\log \sigma \Big( \beta \log \frac{\pi_\theta(\mathbf{y}_w|\mathbf{x})}{\pi_{\text{ref}}(\mathbf{y}_w|\mathbf{x})} \Big) \Big]. \tag{11}$$

As the function $\log \sigma(\cdot)$ is monotonic, its stochastic gradient has the same direction as the SFT loss function, and only differs in learning steps. Notably, with online response data collection ($\mathbf{y}_w$ is generated from the current policy model $\pi_\theta$), this is also known as the Reject Sampling (RS) method (Dong et al., 2023).

**Connection with DPO.**   In equation 8, when $r_w = 0$ and $r_l = -1$, we have $\Delta_r = 1$. The loss function becomes

$$\mathcal{L}(\theta) \;=\; \underset{(\mathbf{y}_w, \mathbf{y}_l, \mathbf{x}) \sim \mathcal{D}}{\mathbb{E}} \Big[ -\log \sigma \Big( \beta \log \frac{\pi_\theta(\mathbf{y}_w|\mathbf{x})}{\pi_{\text{ref}}(\mathbf{y}_w|\mathbf{x})} - \beta \log \frac{\pi_\theta(\mathbf{y}_l|\mathbf{x})}{\pi_{\text{ref}}(\mathbf{y}_l|\mathbf{x})} \Big) \Big], \tag{12}$$

which degenerates to the DPO loss function in equation 3.

Overall, DPO loss function can be taken as *special cases* of MaPPO. In this sense, MaPPO can be taken as a dynamic weighted mechanism, where the weight depends on the relative quality (rewards) of the winning response $\mathbf{y}_w$ and the losing response $\mathbf{y}_l$.

**Gradient Dynamics Analysis.**   To analyze the update of MaPPO, the gradient of equation 8 is

$$\nabla \mathcal{L}_{\text{MaP}}(\theta) \;=\; \underset{(\mathbf{y}_w, \mathbf{y}_l, \mathbf{x}) \sim \mathcal{D}}{\mathbb{E}} \Big[ -\beta(1 - \sigma(u)) \Big( \nabla \log \pi_\theta(\mathbf{y}_w|\mathbf{x}) - \Delta_r \nabla \log \pi_\theta(\mathbf{y}_l|\mathbf{x}) \Big) \Big], \tag{13}$$

where $u = \beta \big( \log \frac{\pi_\theta(\mathbf{y}_w|\mathbf{x})}{\pi_{\text{ref}}(\mathbf{y}_w|\mathbf{x})} - \Delta_r \log \frac{\pi_\theta(\mathbf{y}_l|\mathbf{x})}{\pi_{\text{ref}}(\mathbf{y}_l|\mathbf{x})} \big)$ serves as a confidence measure of preference separation. $1 - \sigma(u)$ down-weights the gradient when the model is already confident in distinguishing $\mathbf{y}_w$ and $\mathbf{y}_l$. $\Delta_r$ directly scales the contribution of the losing sample $\mathbf{y}_l$, modulating the penalization. The gradient norm of MaPPO is upper-bounded compared to DPO, leading to less aggressive updates and more stable policy calibration. We provide a more detailed theoretical analysis, including the stationary convergence analysis in Appendix A.1 and Lipschitz stability analysis in Appendix A.2.

### 4.3   Online MaPPO

Beyond the offline setting, our MaPPO method can be directly used in the online or iterative settings. As shown in Algorithm 1, we describe the online version of MaPPO. In online MaPPO, one key difference is that the responses $\{\mathbf{y}\}$ are generated online from the current policy $\pi_\theta$ instead of the initial policy $\pi_{\theta_0}$ in the offline setting.

In practice, considering training efficiency, online PO can be implemented in an iterative way, known as I-DPO (Dong et al., 2024). In Figure 3, we illustrate the iterative MaPPO pipeline. With a prompt set $\mathcal{D}$, we can equally divide $\mathcal{D}$ into $K$ subsets as $\mathcal{D}_1 \cdots \mathcal{D}_K$. In the $k$-th iteration, we first freeze the current policy model $\pi_\theta$, and then get responses $(\mathbf{y}_1, \mathbf{y}_2)$ from the policy according to the prompt set $\mathcal{D}_k$. We then use a reward model to get the responses' corresponding rewards and collect $(\mathbf{y}_w, \mathbf{y}_l)$ pairs, which reflect the preference. After response collection on $\mathcal{D}_k$, we conduct the MaPPO training process using equation 8 on the subset $\mathcal{D}_k$. After training on the prompt subset, we repeat the process in the next iteration $k + 1$ until we finish all $K$ training iterations.

---

**Algorithm 1** Online MaPPO

---

**Require:** Prompt data set $\mathcal{D}$; Number of iterations $K$; Initial policy model $\theta_0$.
1: **for** $k = 0, \cdots, K-1$ **do**
2:      Sample a prompt $\mathbf{x} \sim \mathcal{D}$.
3:      Sample responses from the current policy $\mathbf{y}_w, \mathbf{y}_l \sim \pi_{\theta_k}(\cdot|\mathbf{x})$.
4:      Get corresponding rewards $r_w \leftarrow r(\mathbf{y}_w, \mathbf{x})$ and $r_l \leftarrow r(\mathbf{y}_l, \mathbf{x})$.
5:      $\Delta_r \leftarrow r(\mathbf{y}_w, \mathbf{x}) - r(\mathbf{y}_l, \mathbf{x})$
6:      Compute $\mathcal{L}(\theta_k)$ according to equation 8.
7:      $\theta_{k+1} \leftarrow \theta_k - \eta \nabla \mathcal{L}(\theta_k)$    # Or other optimizer, e.g., AdamW.
8: **end for**
**Ensure:** $\theta_K$

---

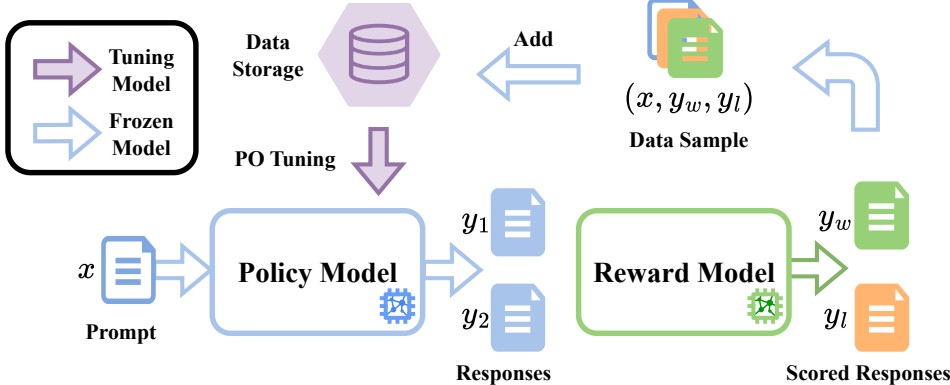

Figure 3: Illustration of the iterative MaPPO pipeline in each iteration $k$.

**Remark.** A reward model (or rule-based verifier) is necessary for all online methods, including online DPO (iterative DPO) (Dong et al., 2024) and reject sampling (Dong et al., 2023).

## 4.4 Adaptation to Other PO Methods

We have shown MaPPO in the offline and online DPO settings. As this Maximum a Posteriori (MaP) method is generally suitable for all DPO variants, we show how MaPPO modifies other DPO variants in this subsection.

Simply replace the MLE part in preference optimization with MaP, and follow the same derivation in Section 4.1. Most DPO variants, as long as MLE is used in the original methods, can be modified with MaPPO as a plugin. We show some widely adopted methods as examples here, including SimPO, IPO, and CPO.

First, in SimPO (Meng et al., 2024), with the length control penalty, the loss function is given as

$$\mathcal{L}_{\text{SimPO}}(\theta) \;=\; \mathop{\mathbb{E}}_{(\mathbf{y}_w, \mathbf{y}_l, \mathbf{x}) \sim \mathcal{D}} \Big[ -\log \sigma \Big( \frac{\beta}{|\mathbf{y}_w|} \log \pi_\theta(\mathbf{y}_w|\mathbf{x}) - \frac{\beta}{|\mathbf{y}_l|} \log \pi_\theta(\mathbf{y}_l|\mathbf{x}) - \gamma \Big) \Big], \tag{14}$$

where $\gamma$ is a constant hyperparameter, $|\mathbf{y}_w|$ and $|\mathbf{y}_l|$ denote the length of $\mathbf{y}_w$ and $\mathbf{y}_l$, respectively.

With the MaPPO plugin, the loss function is modified as

$$\mathcal{L}_{\text{SimPO+}}(\theta) \;=\; \mathop{\mathbb{E}}_{(\mathbf{y}_w, \mathbf{y}_l, \mathbf{x}) \sim \mathcal{D}} \Big[ -\log \sigma \Big( \frac{\beta}{|\mathbf{y}_w|} \log \pi_\theta(\mathbf{y}_w|\mathbf{x}) - \Delta_r \frac{\beta}{|\mathbf{y}_l|} \log \pi_\theta(\mathbf{y}_l|\mathbf{x}) - \gamma \Big) \Big]. \tag{15}$$

In IPO (Azar et al., 2024), the original loss function is

$$\mathcal{L}_{\text{IPO}}(\theta) \;=\; \mathop{\mathbb{E}}_{(\mathbf{y}_w, \mathbf{y}_l, \mathbf{x}) \sim \mathcal{D}} \Big[ \Big( \log \frac{\pi_\theta(\mathbf{y}_w|\mathbf{x})}{\pi_{\text{ref}}(\mathbf{y}_w|\mathbf{x})} - \log \frac{\pi_\theta(\mathbf{y}_l|\mathbf{x})}{\pi_{\text{ref}}(\mathbf{y}_l|\mathbf{x})} - \frac{1}{2\beta} \Big)^2 \Big]. \tag{16}$$

With the MaPPO plugin, the loss function is modified as

$$\mathcal{L}_{\text{IPO+}}(\theta) \;=\; \underset{(\mathbf{y}_w, \mathbf{y}_l, \mathbf{x}) \sim \mathcal{D}}{\mathbb{E}} \left[ \left( \log \frac{\pi_\theta(\mathbf{y}_w|\mathbf{x})}{\pi_{\text{ref}}(\mathbf{y}_w|\mathbf{x})} - \Delta_r \log \frac{\pi_\theta(\mathbf{y}_l|\mathbf{x})}{\pi_{\text{ref}}(\mathbf{y}_l|\mathbf{x})} - \frac{1}{2\beta} \right)^2 \right]. \tag{17}$$

In CPO (Xu et al., 2024), the original loss is

$$\mathcal{L}_{\text{CPO}}(\theta) \;=\; \underset{(\mathbf{y}_w, \mathbf{y}_l, \mathbf{x}) \sim \mathcal{D}}{\mathbb{E}} \left[ -\log \sigma \Big( \beta \log \pi_\theta(\mathbf{y}_w|\mathbf{x}) - \beta \log \pi_\theta(\mathbf{y}_l|\mathbf{x}) \Big) - \lambda \log \pi_\theta(\mathbf{y}_w|\mathbf{x}) \right], \tag{18}$$

where $\lambda$ is a constant hyperparameter.

With the MaPPO plugin, the CPO loss is modified as

$$\mathcal{L}_{\text{CPO+}}(\theta) \;=\; \underset{(\mathbf{y}_w, \mathbf{y}_l, \mathbf{x}) \sim \mathcal{D}}{\mathbb{E}} \left[ -\log \sigma \Big( \beta \log \pi_\theta(\mathbf{y}_w|\mathbf{x}) - \beta \Delta_r \log \pi_\theta(\mathbf{y}_l|\mathbf{x}) \Big) - \lambda \log \pi_\theta(\mathbf{y}_w|\mathbf{x}) \right]. \tag{19}$$

To verify the effectiveness, we show the experimental results with the improvement of these DPO variants in Section 5.3.

**Remark.** With our MaPPO plugin, **no** additional hyperparameter is introduced in all DPO variants.

## 5 Experiments

In this section, we empirically verify the effectiveness of our MaPPO methods.

### 5.1 Setup

**Pipeline Settings.** We follow the RLHF framework in Dong et al. (2024). Instead of costly human annotations, we employ off-the-shelf reward models to generate the preferences. We use the public pre-trained BT reward model[1] as the prior knowledge. For the response selection, we follow the rejection sampling strategy suggested by Liu et al. (2024b); Gulcehre et al. (2023). For each prompt, we generate $n = 8$ responses and use the best-of-8 as $\mathbf{y}_w$ and the worst-of-8 as $\mathbf{y}_l$. We provide hyperparameter details and computing resources in Appendix C.1.

**Dataset.** We use the prompt set in Dong et al. (2024). In the offline setting, we generate responses from the initial model with the whole prompt set. In the online (iterative) setting, we separate the prompt set into three subsets of the same size. The learning process lasts for $K = 3$ iterations. In each iteration, we sample responses from our current policy with one prompt subset, and use preference signals on these responses to improve our policy.

**Models.** To show the scalability of our methods, we choose models in two dimensions: (1) *Model sizes*: Qwen2.5-1.5B-Instruct, Qwen2.5-3B-Instruct, and Qwen2.5-7B-Instruct. (2) *Model series*: Qwen2.5-7B-Instruct, Mistral-7B-Instruct-v0.3, and Llama-3-8B-Instruct in our experiments.

**Evaluation.** We evaluate the model performance on three widely used benchmarks: MT-Bench (Zheng et al., 2023a), AlpacaEval 2.0 (Li et al., 2023), and Arena-Hard v0.1 (Li et al., 2024). MT-Bench contains 80 questions from 8 categories, with answers rated by GPT-4 on a scale of $1-10$. Arena-Hard v0.1 contains 500 technical problem-solving questions, and the answers are compared to reference responses from the baseline model GPT-4-0314. We report the win rate (WR) in percentage as judged by GPT-4 Turbo (Preview-1106). AlpacaEval 2.0 includes 805 questions from five datasets, with the judge model GPT-4 Turbo (Preview-1106) comparing the answers to reference responses from itself. We report the length-controlled (LC) WR as suggested in Dubois et al. (2024).

---

[1] https://huggingface.co/sfairXC/FsfairX-LLaMA3-RM-v0.1

## 5.2 Main Results

Our main results on three standard benchmarks, introduced in Section 5.1, are shown in Table 2. For the alignment methods, we show the evaluation results of Instruction Tuning (IT), the original offline setting (Rafailov et al., 2023) (DPO), and the online setting (Dong et al., 2024) (I-DPO) as described in Section 4.3. For DPO and I-DPO, we show their improvements that incorporate the MaPPO design (+MaPPO).

For Llama-3-8B-Instruct, Mistral-7B-Instruct, and Qwen2.5-7B-Instruct models, the performances are significantly improved with MaPPO on AlpacaEval 2.0 and Arena-Hard in both the offline setting (DPO) and the online setting (I-DPO). It reflects the effectiveness of MaPPO on different model series in both online and offline settings. On the MT-Bench, the performances are slightly improved on Qwen2.5-7B-Instruct and Llama-3-8B-Instruct, because the base models have already achieved very good results on this benchmark, which has limitations to reflect the effective improvement. The improvement on MT-Bench becomes much more significant on models with mediocre base or DPO performances, *e.g.*, Mistral-7B-Instruct and Qwen2.5-1.5B-Instruct. In one model series, the larger models achieve higher overall scores in both base performances and after online & offline alignment tuning, suggesting that scaling up model size enhances alignment capability as expected. With MaPPO, the improvement is consistent in scale with different model sizes in both online and offline settings, and the alignment can make smaller models outperform larger base models.

Table 2: Main evaluation results on three standard benchmarks. ↑ indicates the higher the better.

| Model | Method | AlpacaEval 2.0 ↑ | Arena-Hard ↑ | MT-Bench ↑ |
|---|---|---|---|---|
| Qwen2.5-1.5B-Instruct | IT | 11.10 | 5.0 | 7.06 |
| | DPO | 18.71 | 11.6 | 7.29 |
| | +**MaPPO** | **19.35** +0.64 | **15.3** +3.7 | **7.57** +0.28 |
| | I-DPO | 17.89 | 12.1 | 7.39 |
| | +**MaPPO** | **19.84** +1.95 | **15.7** +3.6 | **7.63** +0.24 |
| Qwen2.5-3B-Instruct | IT | 18.91 | 24.0 | 7.92 |
| | DPO | 20.16 | 29.2 | 8.02 |
| | +**MaPPO** | **26.68** +6.52 | **35.1** +4.9 | **8.13** +0.11 |
| | I-DPO | 19.69 | **36.6** | **8.10** |
| | +**MaPPO** | **25.99** +6.30 | 35.8 −0.8 | 8.01 −0.09 |
| Qwen2.5-7B-Instruct | IT | 27.03 | 42.9 | 8.61 |
| | DPO | 32.01 | 45.5 | 8.56 |
| | +**MaPPO** | **38.24** +6.23 | **59.2** +13.7 | **8.79** +0.23 |
| | I-DPO | 33.80 | 46.9 | **8.55** |
| | +**MaPPO** | **39.10** +5.30 | **61.6** +14.7 | 8.54 −0.01 |
| Mistral-7B-Instruct | IT | 15.35 | 13.1 | 5.40 |
| | DPO | 18.24 | 14.2 | 6.86 |
| | +**MaPPO** | **30.56** +12.32 | **18.4** +4.2 | **7.51** +0.65 |
| | I-DPO | 17.11 | 14.3 | 6.92 |
| | +**MaPPO** | **33.28** +16.14 | **19.6** +5.3 | **7.59** +0.67 |
| Llama-3-8B-Instruct | IT | 10.85 | 10.2 | 7.52 |
| | DPO | 22.48 | 22.4 | 8.07 |
| | +**MaPPO** | **28.37** +5.89 | **29.5** +7.1 | **8.18** +0.11 |
| | I-DPO | 29.47 | 25.6 | 8.01 |
| | +**MaPPO** | **32.68** +3.21 | **31.0** +5.4 | **8.04** +0.03 |

## 5.3 Adaptation to DPO Variants

In Table 3, we show the improvement of the vanilla DPO (Rafailov et al., 2023) and its variants with MaPPO, including widely used I-DPO (Dong et al., 2024), SimPO (Meng et al., 2024), IPO (Gheshlaghi Azar et al., 2024), and CPO (Xu et al., 2024). We list the hyperparameter settings in the DPO variants in Appendix C. Their loss functions with MaPPO adaptation are shown in Section 4.3 and Section 4.4. For all DPO variants, no additional hyperparameter is needed from the MaPPO plugin. For the model in evaluation, we keep Qwen2.5-7B-Instruct as the default model.

In general, MaPPO consistently improves all DPO variants with the MLE design on all three benchmarks. Although it drops a little on MT-Bench in some methods, the original approach has essentially saturated at the achievable score on MT-Bench, which barely reflects the improvement with a variance in evaluation. The overall consistent improvements observed across DPO variants after applying the MaPPO plugin underscore its flexibility and generality in enhancing preference optimization methods. Notably, MaPPO effectively complements both simple and complex variants without requiring architectural modifications or hyperparameter tuning. For instance, SimPO benefits from the MaPPO adjustment by further balancing the length-controlled optimization with better calibration of confidence scores, while IPO and CPO experience gains due to MaPPO's capacity to regularize reward signals with prior knowledge, mitigating overfitting to pairwise preferences. The improvements span diverse evaluation metrics and benchmarks, demonstrating that MaPPO's reward-aware calibration systematically addresses the shortcomings of MLE-based objectives inherent in existing variants. This indicates that MaPPO is not merely a tweak, but a general principle that can be seamlessly integrated into the PO pipelines to achieve more reliable alignment results.

Table 3: Evaluation results of DPO variants with a MaPPO plugin. ↑ indicates the higher the better.

| Method | AlpacaEval 2.0 ↑ | Arena-Hard ↑ | MT-Bench ↑ |
|---|---|---|---|
| DPO (Rafailov et al., 2023) | 32.01 | 45.5 | 8.56 |
| **+MaPPO** | **38.24** +6.23 | **59.2** +13.7 | **8.79** +0.23 |
| I-DPO (Dong et al., 2024) | 33.80 | 46.9 | **8.55** |
| **+MaPPO** | **39.10** +5.30 | **61.6** +14.7 | 8.54 −0.01 |
| SimPO (Meng et al., 2024) | 25.15 | 64.2 | **9.02** |
| **+MaPPO** | **32.75** +7.60 | **69.5** +5.3 | 8.94 −0.08 |
| IPO (Gheshlaghi Azar et al., 2024) | 27.76 | 53.0 | 8.83 |
| **+MaPPO** | **28.84** +1.08 | **64.4** +11.4 | **8.84** +0.01 |
| CPO (Xu et al., 2024) | 32.94 | 47.6 | 8.62 |
| **+MaPPO** | **33.71** +0.77 | **54.1** +6.5 | **8.68** +0.06 |

## 5.4 Other Results on Academic Benchmarks

It is widely observed that alignment impairs models' performance on calibration, reasoning, and accuracy (Ouyang et al., 2022; Lin et al., 2024; Zhang et al., 2025), which is also known as the alignment tax. As a result, it is also needed to assess the model's performance using more academic benchmarks. In this subsection, we investigate whether the serval methods for alignment with human preference could sacrifice the general model performance.

We test the performance on six widely used academic benchmarks, evaluating various model abilities, including explicit instruction following (IFEval) (Zhou et al., 2023), general knowledge (GPQA) (Rein et al., 2024), multitask language understanding (MMLU) (Hendrycks et al., 2021), commonsense reasoning (HellaSwag) (Zellers et al., 2019), human falsehoods mimicking (TruthfulQA) (Lin et al., 2022), and math word problem-solving (GSM8K) (Cobbe et al., 2021).

We show the results on the six academic benchmarks from Llama-3-8B-Instruct model in Table 4, and Qwen2.5-7B-Instruct model in Table 5. In general, for offline DPO with MaPPO outperforms the original DPO in all benchmarks for both models. The improvement is significant on GSM8K for Qwen2.5-7B-Instruct, and on TruthfulQA for Llama-3-8B-Instruct. For the iterative DPO with MaPPO, the performances are

better than the original I-DPO on most benchmarks, and maintains the performances on IFEval and GPQA. Overall, the performances of online methods are better than offline methods, and MaPPO generally improves or maintains the performances on academic benchmarks in both settings.

Table 4: Evaluation results on six academic benchmarks with Llama-3-8B-Instruct model.

| Method | IFEval ↑ | GPQA ↑ | MMLU ↑ | HellaSwag ↑ | TruthfulQA ↑ | GSM8K ↑ |
|---|---|---|---|---|---|---|
| IT | 70.4 | 30.2 | 62.4 | 78.6 | 53.7 | 73.4 |
| DPO | 77.0 | 27.5 | 62.7 | 79.5 | 51.5 | 75.5 |
| +**MaPPO** | **82.0** | **29.5** | **63.2** | **80.1** | **58.2** | **79.5** |
| I-DPO | 74.6 | **29.8** | 63.1 | 80.5 | 60.7 | 81.3 |
| +**MaPPO** | **76.4** | 28.8 | **63.5** | **80.7** | **63.7** | **82.4** |

Table 5: Evaluation results on six academic benchmarks with Qwen2.5-7B-Instruct model.

| Method | IFEval ↑ | GPQA ↑ | MMLU ↑ | HellaSwag ↑ | TruthfulQA ↑ | GSM8K ↑ |
|---|---|---|---|---|---|---|
| IT | 73.5 | 31.5 | 71.8 | 62.1 | 56.4 | 81.7 |
| DPO | 73.2 | 32.0 | 71.9 | 62.0 | 57.1 | 71.3 |
| +**MaPPO** | **73.8** | **33.1** | **72.0** | **62.1** | **59.2** | **80.1** |
| I-DPO | **72.9** | 33.0 | 71.9 | 62.2 | 55.9 | 73.2 |
| +**MaPPO** | 72.6 | **33.3** | **72.9** | **62.2** | **56.2** | **82.0** |

## 6 Discussions

**Limitations and Future Work.**
1. Our results indicate that larger models consistently perform better with the MaPPO method. Future work with more computing resources could explore applying the proposed training pipeline to models larger than 8B parameters.

2. The prior knowledge function design relies on experts' domain knowledge. We give a reasonable and general design. Future works could explore different designs in specific domains.

3. If resources permit, human evaluation can be included for better judgment of alignment.

**Conclusion.**  In this work, we propose MaPPO, a general and principled framework for preference optimization that incorporates prior knowledge into the optimization objective. By extending the Maximum Likelihood Estimation (MLE)-based Preference Optimization (PO) approach to a Maximum a Posteriori (MaP) formulation, MaPPO effectively mitigates confidence degeneration and provides a more calibrated training signal. Our method requires no additional hyperparameters, supports both offline and online settings, and can be seamlessly integrated into existing Direct Preference Optimization (DPO) variants, including widely used SimPO, IPO, and CPO. Without sacrificing efficiency, extensive empirical results demonstrate that MaPPO consistently improves alignment performance on different model series (e.g., Qwen, Mistral, and Llama), and on scaling to different model sizes (e.g., 1.5B, 3B, 7B, and 8B) across three standard benchmarks, including MT-Bench, AlpacaEval 2.0, and Arena-Hard. We also evaluate the performance on six widely used academic benchmarks after alignment, which shows that MaPPO, compared to the other methods, maintains the performance in various dimensions.

**Broader Impact Statement.**  MaPPO aligns language models by incorporating prior reward knowledge into preference optimization, leading to better-calibrated and more robust outputs. While beneficial, it relies on reward models that may encode biases or misrepresent human values, potentially reinforcing harmful patterns. Its use in persuasive or deceptive applications also poses potential impacts. To mitigate these impacts, we encourage careful curation and auditing of reward models, broader involvement in defining reward signals, and transparency in how preference optimization frameworks, such as MaPPO, are applied in real-world AI systems.

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

# A Theoretical Results

## A.1 Stationary Convergence of MaPPO

Recall that

$$
\begin{aligned}
\mathcal{L}_{\mathrm{MaP}}(\theta) &= \mathop{\mathbb{E}}_{(\mathbf{y}_w, \mathbf{y}_l, \mathbf{x}) \sim \mathcal{D}} \left[ -\log \sigma \left( \beta \log \frac{\pi_\theta(\mathbf{y}_w|\mathbf{x})}{\pi_{\mathrm{ref}}(\mathbf{y}_w|\mathbf{x})} - \Delta_r \beta \log \frac{\pi_\theta(\mathbf{y}_l|\mathbf{x})}{\pi_{\mathrm{ref}}(\mathbf{y}_l|\mathbf{x})} \right) \right] \\
&= \mathop{\mathbb{E}}_{(\mathbf{y}_w, \mathbf{y}_l, \mathbf{x}) \sim \mathcal{D}} \left[ -\log \sigma(u) \right].
\end{aligned}
\tag{20}
$$

At the first-order stationary point (FOSP), the gradient of the loss with respect to $\theta$ becomes 0. Thus, we have

$$
\nabla \mathcal{L}_{\mathrm{MaP}}(\theta) = \mathop{\mathbb{E}}_{(\mathbf{y}_w, \mathbf{y}_l, \mathbf{x}) \sim \mathcal{D}} \left[ -\beta(1 - \sigma(u)) \left( \nabla \log \pi_\theta(\mathbf{y}_w|\mathbf{x}) - \Delta_r \nabla \log \pi_\theta(\mathbf{y}_l|\mathbf{x}) \right) \right] = 0.
\tag{21}
$$

Let the optimal policy be $\pi_\star$. The above equation holds when

$$
\nabla \log \pi_\star(\mathbf{y}_w|\mathbf{x}) - \Delta_r \nabla \log \pi_\star(\mathbf{y}_l|\mathbf{x}) = 0.
\tag{22}
$$

Thus, the optimal policy achieves

$$
\log \pi_\star(\mathbf{y}_w|\mathbf{x}) = \Delta_r \log \pi_\star(\mathbf{y}_l|\mathbf{x}) + c,
\tag{23}
$$

where $c$ is a scaling constant determined by the initialization. Thus, the model learns a stable log-linear relationship between preferred and less-preferred responses, scaled by the prior reward gap.

In DPO, the optimal policy at the FOSP is

$$
\log \pi_\star(\mathbf{y}_w|\mathbf{x}) = \log \pi_\star(\mathbf{y}_l|\mathbf{x}) + c.
\tag{24}
$$

DPO converges to maximizing the log-odds between $\mathbf{y}_w$ and $\mathbf{y}_l$, but no inherent bound on the preference gap, which can lead to confidence degeneration. As training progresses, DPO may tend to decrease the likelihood of both $\mathbf{y}_w$ and $\mathbf{y}_l$ (the *squeezing effect*), because there is no constraint on absolute probabilities – only the relative gap matters.

MaPPO prevents overconfidence and instability by grounding optimization in the reward-based prior. The FOSP of MaPPO guarantees a bounded, calibrated log-probability ratio between $\mathbf{y}_w$ and $\mathbf{y}_l$. Naturally limits the *squeezing effect* by scaling the impact of $\mathbf{y}_l$ via $\Delta_r$. Therefore, MaPPO is theoretically more stable, especially for near-tie preference pairs and in large models where DPO can exacerbate miscalibration.

## A.2 Lipschitz Stability

First, we list the standard assumptions for the analysis.

**Assumption A.1.**

1. The score function is Lipschitz continuous as $\|\nabla \log \pi_\theta(\mathbf{y}|\mathbf{x}) - \nabla \log \pi_{\theta'}(\mathbf{y}|\mathbf{x})\| \leq M_g$.

Let the gradient operator be defined as

$$
\tau_\theta := \nabla \mathcal{L}_{\mathrm{MaP}}(\theta).
\tag{25}
$$

Then, the gradient operator $\tau$ is Lipschitz continuous with

$$
\|\tau_\theta - \tau_{\theta'}\| \leq L_{\mathrm{MaP}} \|\theta - \theta'\|,
\tag{26}
$$

where $L_{\mathrm{MaP}} = \beta(1 - \sigma(u))(1 + \Delta_r) M_g < \beta(1 + \Delta_r) M_g$.

*Proof.* We have

$$\tau_\theta = -\beta(1-\sigma(u))\Big(\nabla \log \pi_\theta(\mathbf{y}_w|\mathbf{x}) - \Delta_r \nabla \log \pi_\theta(\mathbf{y}_l|\mathbf{x})\Big). \tag{27}$$

The norm of the gradient difference is

$$
\begin{aligned}
\|\tau_\theta - \tau_{\theta'}\| &= \beta(1-\sigma(u)) \cdot \|\nabla \log \pi_\theta(\mathbf{y}_w|\mathbf{x}) - \nabla \log \pi_{\theta'}(\mathbf{y}_w|\mathbf{x})\| \\
&\quad + \beta(1-\sigma(u))\Delta_r \cdot \|\nabla \log \pi_\theta(\mathbf{y}_l|\mathbf{x}) - \nabla \log \pi_{\theta'}(\mathbf{y}_l|\mathbf{x})\|.
\end{aligned}
\tag{28}
$$

We have that $\sigma(u)$ is Lipschitz with constant $\frac{1}{4}$. Combining the $M_g$ Lipschitz of the score function $\nabla \log \pi_\theta(\mathbf{y}|\mathbf{x})$, we have

$$\|\tau_\theta - \tau_{\theta'}\| \le \beta(1-\sigma(u))(1+\Delta_r)M_g\|\theta-\theta'\| \;\coloneqq\; L_{\mathrm{MaP}}\|\theta-\theta'\|. \tag{29}$$

$\square$

In contrast, in DPO, the gradient is Lipschitz continuous as

$$\|\tau_\theta - \tau_{\theta'}\| \le L_{\mathrm{DPO}}\|\theta-\theta'\|, \tag{30}$$

where $L_{\mathrm{DPO}} = 2\beta(1-\sigma(u))M_g < 2\beta M_g$. The Lipschitz constant of the gradients in DPO is larger than that in MaPPO, which shows the gradients have less stability in DPO.

With a formal upper bound on MaPPO's gradient variation, we show a clear theoretical justification for why MaPPO is less prone to instability and exploding confidence gaps, compared to DPO.

# B    Supplementary Examples

> **Prompt x**
>
> Solve the following math problem step-by-step. Simplify your answer as much as possible. Present your final answer as $\boxed{Your Answer}$. If the sum of two positive integers is 22 and the difference of their squares is 44, what is the product of the two integers?

## B.1    Before Preference Optimization

> **Response $\mathbf{y}_w$, $r = 0.93$**
>
> To solve this problem step-by-step, let's denote the two positive integers as $x$ and $y$ with $x > y$.
> Step 1: Set Up Equations
>    Given:
>        1. The sum of the two integers: $x + y = 22$
>        2. The difference of their squares: $x^2 - y^2 = 44$
> Step 2: Factorize the Difference of Squares Recall that the difference of squares can be factored as:
>
> $$x^2 - y^2 = (x + y)(x - y)$$
>
> Substitute the given sum $x + y = 22$ into the factored form:
>
> $$44 = 22(x - y)$$
>
> Step 3: Solve for $x - y$ Divide both sides by 22:
>
> $$x - y = 2$$
>
> Step 4: Solve the System of Linear Equations Now, we have the system: 1. $x + y = 22$ 2. $x - y = 2$
> Add these two equations to eliminate $y$:
>
> $$(x + y) + (x - y) = 22 + 2$$
>
> $$2x = 24$$
>
> $$x = 12$$
>
> Substitute $x = 12$ back into the first equation to find $y$:
>
> $$12 + y = 22$$
>
> $$y = 10$$
>
> Step 5: Calculate the Product The product of the two integers $x$ and $y$ is:
>
> $$x \cdot y = 12 \cdot 10 = 120$$
>
> Final Answer
>
> $$\boxed{120}$$

**Response $\mathbf{y}_l$, $r = 0.90$**

To solve this problem step-by-step, let's denote the two positive integers as $x$ and $y$, where $x > y$. We are given two conditions:

1. The sum of the two integers is 22:

$$x + y = 22$$

2. The difference of their squares is 44:

$$x^2 - y^2 = 44$$

We can use the identity for the difference of squares, $x^2 - y^2 = (x + y)(x - y)$, to rewrite the second equation. Substituting $x + y = 22$ into the identity gives:

$$(x + y)(x - y) = 44 \implies 22(x - y) = 44$$

Dividing both sides by 22:

$$x - y = 2$$

Now we have a system of linear equations:

$$\begin{cases} x + y = 22 \\ x - y = 2 \end{cases}$$

Adding these two equations eliminates $y$:

$$(x + y) + (x - y) = 22 + 2 \implies 2x = 24 \implies x = 12$$

Substituting $x = 12$ back into the first equation to solve for $y$:

$$12 + y = 22 \implies y = 10$$

Thus, the two integers are $x = 12$ and $y = 10$.
To find the product of the two integers:

$$xy = 12 \times 10 = 120$$

Therefore, the product of the two integers is $\boxed{120}$.

Figure 4: Before MLE optimization, the model consistently generates high-quality (high rewards) answers $\mathbf{y}_w$ and $\mathbf{y}_l$ with prompt $\mathbf{x}$.

## B.2 After MLE Optimization

> **Response $\mathbf{y}_w$, $r = 0.87$**
>
> To solve the given problem, we'll proceed through the following steps systematically:
> Given:
>     1. The sum of two positive integers $x$ and $y$ is 22.
>     2. The difference of their squares is 44.
> We need to determine the product of these two integers.
> Step-by-Step Solution:
> 1. Setting up Equations:
> - From the first condition:
> $$x + y = 22$$
> - From the second condition (expressing the difference of squares):
> $$x^2 - y^2 = 44$$
> We know that $x^2 - y^2$ can be factored using the difference of squares formula:
> $$x^2 - y^2 = (x + y)(x - y)$$
> 2. Substitute $x + y$:
> - Using the known value from equation (1) in the factored form of the difference of squares:
> $$x^2 - y^2 = (x + y)(x - y) \implies 44 = 22(x - y)$$
> - Solve for $x - y$:
> $$22(x - y) = 44 \implies x - y = 2$$
> 3. Solving the System of Linear Equations:
> - Now, we have the system of equations:
> $$x + y = 22$$
> $$x - y = 2$$
> - Add these two equations to eliminate $y$:
> $$(x + y) + (x - y) = 22 + 2$$
> $$2x = 24 \implies x = 12$$
> - Substitute $x = 12$ back into $x + y = 22$ to find $y$:
> $$12 + y = 22 \implies y = 10$$
> 4. Finding the Product:
> - With $x = 12$ and $y = 10$, calculate the product:
> $$xy = 12 \times 10 = 120$$
> Thus, the product of the two integers is $\boxed{120}$.

---

**Response $\mathbf{y}_l$, $r = 0.43$**

Step 1: Interpret the problem and set up equations based on the given information. Given that the sum of two positive integers is 22 and the difference of their squares is 44, we need to find these integers and subsequently their product.

Let's denote the two integers as $a$ and $b$ such that $a > b$, to simplify handling the difference and sum relationships given directly without specifying which integer represents $a$ or $b$ in terms of being greater than the other directly addresses addressing conditions related directly to interpreting the context regarding how the operations (sum, difference of squares) relate between two specified integers, based on the given descriptions for solving in the context presented, focusing on correctly interpreting and directly addressing the specific requirements as outlined for accurately proceeding with solving based on the given details, to correctly proceed according to provided instructions and the nature of solving based on the specifications given for dealing with the described conditions around the operation outcomes involving the integers in question.

Step 2: Translate the given conditions into mathematical equations.
- The sum of the two integers equals 22: $a + b = 22$.
- The difference of their squares equals 44: $a^2 - b^2 = 44$.

Step 3: Utilize algebraic identities for simplification. From the second equation, apply the algebraic identity for the difference of squares:

$$a^2 - b^2 = (a + b)(a - b)$$

Thus, substituting in the known value for $a + b$:

$$(a + b)(a - b) = 44 \implies 22(a - b) = 44.$$

Step 4: Solve for $a - b$. Divide both sides by 22:

$$a - b = 2.$$

Step 5: Solve the system of linear equations. We now have two simultaneous equations:
1. $a + b = 22$
2. $a - b = 2$
Add the two equations together to eliminate $b$:

$$(a + b) + (a - b) = 22 + 2 \implies 2a = 24 \implies a = 12.$$

Substitute $a = 12$ back into the first equation to solve for $b$:

$$12 + b = 22 \implies b = 10.$$

Step 6: Find the product of $a$ and $b$. The product of the two integers $a$ and $b$ is:

$$ab = 12 \times 10 = 120.$$

Step 7: Box the final answer.

$$\boxed{120}$$

Figure 5: After MLE optimization, the model degenerates, and the outputs $\mathbf{y}_w$ and $\mathbf{y}_l$ become verbose (low rewards) with prompt $\mathbf{x}$.

## B.3  After MaP Optimization

---

**Response $\mathbf{y}_w$, $r = 0.94$**

To solve this problem step-by-step, let's denote the two positive integers by $x$ and $y$, where without loss of generality, we assume $x > y$.

Given:

1. The sum of the two integers: $x + y = 22$
2. The difference of their squares: $x^2 - y^2 = 44$

Step 1: Express the Difference of Squares
 We know from algebra that:
$$x^2 - y^2 = (x + y)(x - y)$$

Given $x + y = 22$, we substitute this into the equation:

$$44 = 22(x - y)$$

Step 2: Solve for $x - y$
 Divide both sides by 22:
$$x - y = 2$$

Step 3: Solve the System of Linear Equations
 Now, we have a system of linear equations:
  1. $x + y = 22$
  2. $x - y = 2$
 Add these two equations to eliminate $y$:

$$(x + y) + (x - y) = 22 + 2$$

$$2x = 24$$

$$x = 12$$

Subtract the second equation from the first to eliminate $x$:

$$(x + y) - (x - y) = 22 - 2$$

$$2y = 20$$

$$y = 10$$

Step 4: Calculate the Product of the Two Integers
 Now that we have $x = 12$ and $y = 10$, we find their product:

$$xy = 12 \times 10 = 120$$

Thus, the product of the two integers is:

$$\boxed{120}$$

---

**Response $\mathbf{y}_l$, $r = 0.88$**

To solve this problem, we'll use algebraic methods to represent the given conditions and then solve for the integers. Here are the steps:

1. Define Variables:

   Let the two positive integers be $x$ and $y$, with $x > y$.

2. Set Up Equations Based on Given Information:

   - The sum of the two integers is 22:

   $$x + y = 22$$

   - The difference of their squares is 44:

   $$x^2 - y^2 = 44$$

3. Factor the Difference of Squares:

   Recall that $x^2 - y^2$ can be factored as $(x + y)(x - y)$. Therefore:

   $$(x + y)(x - y) = 44$$

   Substitute $x + y = 22$ into the equation:

   $$22(x - y) = 44$$

4. Solve for $x - y$:

   Divide both sides by 22:

   $$x - y = 2$$

5. Set Up a System of Linear Equations:

   We now have the system:

   $$\begin{cases} x + y = 22 \\ x - y = 2 \end{cases}$$

6. Add the Two Equations to Solve for $x$:

   Adding the equations eliminates $y$:

   $$(x + y) + (x - y) = 22 + 2$$

   Simplifies to:

   $$2x = 24 \implies x = 12$$

7. Substitute $x = 12$ Back to Find $y$:

   Using $x + y = 22$:

   $$12 + y = 22 \implies y = 10$$

8. Calculate the Product of the Two Integers:

   The product of $x$ and $y$ is:

   $$x \cdot y = 12 \cdot 10 = 120$$

The answer is

$$\boxed{120}$$

Figure 6: After MaP optimization, the model consistently generates high quality outputs with prompt $\mathbf{x}$.

## C   Supplementary Experiments

### C.1   Supplementary Experimental Settings

**Hyperparameter settings.**   We follow the standard settings and list the hyperparameter details in the training process of MaPPO in Table 6. We keep the hyperparameter settings for different model series and model sizes, including Qwen2.5-{1.5B, 3B, 7B}-Instruct, Mistral-7B-Instruct, and Llama-3-8B-Instruct models.

Table 6: Hyperparameter settings in MaPPO.

| Hyperparameter | Value |
| --- | --- |
| global batch size | 128 |
| learning rate $\eta$ | $5 \times 10^{-7}$ |
| warmup steps | 100 |
| weight decay | 0.01 |
| optimizer | AdamW |
| KL weight $\beta$ | 0.1 |
| number of responses $n$ | 8 |
| temperature $T$ | 1.0 |
| precision | `bfloat16` |

Table 7: Hyperparameter settings in DPO Variants.

| Hyperparameter | Value |
| --- | --- |
| SimPO: $\gamma$ | 1 |
| IPO: $\beta$ | 0.1 |
| CPO: $\lambda$ | 0.2 |

In Table 7, we list the extra hyperparameters in the reproduce of the DPO variants. The other hyperparameter settings keep the same in Table 6. Notably, we choose nearly the best hyperparameters for the other methods, and our reproduction achieves higher performances than the original or other reproduction reports on some benchmarks, *e.g.*, SimPO on Arena-Hard.

**Computing Resources.**   All tasks are trained and evaluated on a platform with 8 NVIDIA A100 GPUs on each node, and 80 GB of memory for each GPU. Each training task requires between 4 and 20 hours to execute, depending on the size of the model.

### C.2   Supplementary Results

**Ablation Study on RMs.**   As illustrated in Eq. 5, the prior is instantiated via a reward model (RM). To better understand the role of RMs, we conduct an ablation study by integrating different open-source RMs into our framework, including BT model[2], MoE model[3], and uncertainty-aware model[4]. As shown in Table 8, the absolute performance of I-DPO varies considerably depending on the chosen RM. Importantly, across all settings, our proposed MaPPO plugin consistently boosts performance on AlpacaEval 2.0, Arena-Hard, and MT-Bench. These results highlight two key insights: (i) the choice of RM can influence the strength of supervision and the final outcomes, and (ii) MaPPO is robust to such variation, reliably enhancing DPO regardless of the underlying RM. This robustness underscores the generality and flexibility of our framework.

---

[2]https://huggingface.co/sfairXC/FsfairX-LLaMA3-RM-v0.1
[3]https://huggingface.co/RLHFlow/ArmoRM-Llama3-8B-v0.1
[4]https://huggingface.co/LxzGordon/URM-LLaMa-3.1-8B

Table 8: Ablation on reward models. ↑ indicates the higher the better.

| Model | Reward Model | Method | AlpacaEval 2.0 ↑ | Arena-Hard ↑ | MT-Bench ↑ |
|---|---|---|---|---|---|
| Qwen2.5-7B-IT | sfairXC/FsfairX-LLaMA3-RM-v0.1 | IT | 27.03 | 42.9 | 8.61 |
| | | I-DPO | 33.80 | 46.9 | **8.55** |
| | | **+MaPPO** | **39.10** +5.30 | **61.6** +14.7 | 8.54 −0.01 |
| | RLHFlow/ArmoRM-Llama3-8B-v0.1 | IT | 27.03 | 42.9 | 8.61 |
| | | I-DPO | 28.79 | 48.3 | 8.58 |
| | | **+MaPPO** | **31.56** +2.77 | **52.0** +3.7 | **8.60** +0.02 |
| | LxzGordon/URM-LLaMa-3.1-8B | IT | 27.03 | 42.9 | 8.61 |
| | | I-DPO | 32.26 | 55.1 | 8.60 |
| | | **+MaPPO** | **36.17** +3.91 | **59.4** +4.3 | **8.62** +0.02 |
| Llama-3-8B-IT | sfairXC/FsfairX-LLaMA3-RM-v0.1 | IT | 10.85 | 10.2 | 7.52 |
| | | I-DPO | 29.47 | 25.6 | 8.01 |
| | | **+MaPPO** | **32.68** +3.21 | **31.0** +5.4 | **8.04** +0.03 |
| | RLHFlow/ArmoRM-Llama3-8B-v0.1 | IT | 10.85 | 10.2 | 7.52 |
| | | I-DPO | 12.69 | 8.1 | 7.44 |
| | | **+MaPPO** | **16.69** +4.00 | **16.4** +8.3 | **7.78** +0.34 |
| | LxzGordon/URM-LLaMa-3.1-8B | IT | 10.85 | 10.2 | 7.52 |
| | | I-DPO | 25.61 | 30.4 | 8.06 |
| | | **+MaPPO** | **27.83** +2.22 | **33.7** +3.3 | **8.10** +0.04 |

Table 9: Comparison w/wo SFT-B and w/wo MaPPO. ↑ indicates the higher the better.

| Model | Method | AlpacaEval 2.0 ↑ | Arena-Hard ↑ | MT-Bench ↑ |
|---|---|---|---|---|
| Qwen2.5-7B-IT | IT | 27.03 | 42.9 | 8.61 |
| | I-DPO | 33.80 | 46.9 | **8.55** |
| | **+MaPPO** | **39.10** +5.30 +5.30 | **61.6** +14.7 +14.7 | 8.54 −0.01 −0.01 |
| | **+SFT-B** | 51.36 +17.56 | 33.5 −13.4 | 8.01 −0.54 |
| | **+SFT-B +MaPPO** | **55.84** +22.04 +4.48 | **43.5** −3.4 +10.0 | **8.40** −0.15 +0.39 |
| Llama-3-8B-IT | IT | 10.85 | 10.2 | 7.52 |
| | I-DPO | 29.47 | 25.6 | 8.01 |
| | **+MaPPO** | **32.68** +3.21 +3.21 | **31.0** +5.4 +5.4 | **8.04** +0.03 +0.03 |
| | **+SFT-B** | 35.55 +6.08 | 30.1 +4.5 | 8.02 +0.01 |
| | **+SFT-B +MaPPO** | **36.21** +6.74 +0.66 | **31.5** +5.9 +1.4 | **8.03** +0.02 +0.01 |

**Comparison with Other Methods.** We also compare our approach with complementary techniques, such as the method proposed by Ren & Sutherland (2024), which introduces an additional SFT-based pre-processing (denoted as SFT-B) to mitigate the squeezing effect. It is worthy to mention that our proposed method is compatible with SFT-B.

In Table 9, we use the blue color to indicate the improvement with the MaPPO plugin, and the green color to indicate the improvement compared to I-DPO. We find SFT-B alone can bring substantial gains in certain metrics but may also introduce trade-offs on others (e.g., lower Arena-Hard performance on Qwen2.5-7B). By contrast, when combined with MaPPO, the two methods operate jointly and achieve consistent improvements across most benchmarks. These findings demonstrate that MaPPO can be seamlessly integrated with diverse training strategies, further broadening its applicability in practice.

# D  Further Discussions

## D.1  Prior Function

In Bayes estimation, the prior distribution is usually constructed by experts with domain knowledge without an exception to avoid it (Jaynes, 2007). We choose a simple form in equation 5, which has the same structure

as the widely used prior function (Blei et al., 2010) and aligns with the softmax probability. This brings a very clean result in equation 8. In this paper, we offer an effective function with good performances and concise formulation. We bring the MaP design into the DPO pipeline, and the prior function construction is not the aim of this paper. Other function choices that are designed by domain experts are also acceptable and open to be used.

On the other hand, we perform ablation studies on reward models in Appendix C.2, which shows the robustness to different reward signals.

## D.2    Recent RL Methods

In addition to the methods discussed in the introduction section, several recent variants of RL methods for LLMs further contextualize the landscape in which PO operates. DAPO (Yu et al., 2025) introduces a scalable RL system that integrates data selection, clip-higher, overlong reward shaping, and token-level loss into a unified framework, demonstrating strong empirical performance at scale. GSPO (Zheng et al., 2025) generalizes sequence-level policy optimization to groupwise preference structures, enabling the use of ranked or collectively generated signals and thereby broadening the applicability of RL methods into mixture-of-experts (MoE) architectures. PRO (Guo et al., 2025) proposes a proximalized formulation of preference optimization that decomposes preference signals into stable update components, allowing the method to accommodate diverse feedback modalities while improving optimization stability. Together, these approaches represent the most recent advances in preference optimization and RL post-training, situating MaPPO alongside them highlights its complementary role as a principled, prior-informed extension that remains compatible with the post-training pipelines. An interesting direction for future work is exploring how prior knowledge can be integrated into these RL methods.

