# OpenReview forum: "MaPPO: Maximum a Posteriori Preference Optimization with Prior Knowledge"
_TMLR — Rejected by TMLR_

### Review · Reviewer_wQhG · 2025-08-25

**Summary Of Contributions:**

The paper introduces MaP Preference Optimization for aligning LLMs with human preference. Unlike DPO, MaPPo adopts a MaP formulation to explicitly incorporate prior reward knowledge into the optimization objective. This approach could mitigate the "squeezing effect", providing more calibrated training without introducing additional hyperparameters. The authors demonstrate the effectiveness across multiple on extensive benchmarks.

**Audience:**

Yes

**Audience Explanation:**

1. Preference optimization is an area of interest in optimization, especially for llm training.
2. The authors propose to consider the DPO into MaP is interesting, and could be a possible solution for alleviate the "squeezing effect".
3. The proposed prior can work in cooperation with DPO and its variants, which is an advantage.

**Broader Impact Concerns:**

The works have included a discussion regarding the border impact concerns. In my opinion, no further concerns are needed.

**Claims And Evidence:**

Yes

**Claims Explanation:**

1. Improving MLE to MaP allows the incorporation of prior knowledge, which can be particularly beneficial for reward learning. The authors clearly demonstrate its connections to both DPO and SFT.
2. The authors provide extensive experiments across multiple LLM families (Qwen, Mistral, Llama) and sizes (1.5B–8B) on benchmarks such as MT-Bench, AlpacaEval 2.0, Arena-Hard, and six additional academic benchmarks to demonstrate the effectiveness of the proposed method.

**Requested Changes:**

1. Although the authors provide an elegant prior, how to set the prior properly remains a problem in MaP, and the paper does not offer sufficient insights.

2. In my opinion, the authors should include studies on absolute response confidence in the experimental section, particularly with comparisons to related methods, such as the approach used in [1].

    [1] Ren, Yi, and Danica J. Sutherland. Learning Dynamics of LLM Finetuning. ICLR 2025.

3. Although Eq. 9 could be interpreted as a variant of SFT, removing the sigmoid function may lead to negative inputs for the logarithm, which is problematic. The authors should demonstrate in a more rigorous manner.

---

### Review · Reviewer_U27W · 2025-08-29

**Summary Of Contributions:**

This paper proposes Maximum a Posteriori Preference Optimization (MaPPO). Compared to the standard preference learning framework, such as DPO, that only uses maximum likelihood estimation (MLE) as the loss, this paper additionally considers prior reward knowledge as a prior-informed regularization term. This is also the major contribution of this paper. The authors also validate this idea using theoretical guarantees and empirical evaluations.

The strengths of this paper are:
1. a novel idea of augmenting DPO with prior reward knowledge. I have never seen this before.
2. rigorous theoretical guarantee of the proposed algorithm.
3. abundant empirical evaluations on various domains (MT-bench, AlpacaEval, Arena-Hard v0.1) and models (Qwen, Mistral, Llama).

The weaknesses of this paper are: I think an ablation on the prior reward quality can make the paper stronger. While the paper claims that a simple reward prior is already effective, I am curious that whether a better reward prior can lead to a stronger improvement. Or a discussion on this matter can be interesting.

**Audience:**

Yes

**Audience Explanation:**

I think the idea of incorporating prior reward signal in the objective function is interesting and opens the door to motivate future researchers to work on this topic. In addition, the new objective does not introuce new hyperparameters compared to the original MLE-based DPO loss, which improves the flexibility and efficiency of this method.

**Broader Impact Concerns:**

The paper has a Broader Impact Statement, which I think is reasonable.

**Claims And Evidence:**

Yes

**Claims Explanation:**

The effectiveness of the proposed method is suported by the rigorous theoretical guarantee and empirical evaluations. I check the proof in Appendix A and I do not find issues.

**Requested Changes:**

In general, I think this paper is already well-formatted. The key idea is clear and the logic is easy to follow. The proposed loss function and algorithm are well-supported by both theoretical guarantee and empirical evaluations. Therefore, I recommend acceptance .

As mentioned in the summary, there is a way to make this paper stronger, which is to add some discussions on how the quality of the prior reward affects the algorithm performance. However, this does not affect my recommendation to this paper and is just a plus that makes the paper stronger.

---

> ### Comment · Reviewer_U27W · 2025-12-19
>
> Thanks for the additional ablation results. I'm very happy with the revision.

---

### Review · Reviewer_2SVh · 2025-12-04

**Summary Of Contributions:**

This paper introduces MaPPO (Maximum-a-Posteriori Preference Optimization), a principled and lightweight extension of Direct Preference Optimization (DPO) that integrates data-driven prior reward knowledge into the preference-learning objective. While prior DPO-style algorithms fundamentally rely on a Maximum Likelihood Estimation(MLE) view of preference learning, MaPPO reframes the objective as Maximum-a-Posteriori (MaP) estimation by injecting a log-prior regularizer weighted by reward gaps, thereby addressing key limitations of standard DPO.

The paper’s contributions are clear:

1. Theoretical Framework:
The authors derive MaPPO directly from Bradley–Terry preference modeling, showing that incorporating a prior term yields a modified logistic loss where the losing response is weighted by the reward gap
$\Delta r = r_w - r_l$.
They show:
- DPO is a special case when $\Delta r = 1$.
- SFT (or RS in online setting) is recovered when $\Delta r = 0$.
- The gradient update becomes more stable and globally calibrated.
Detailed gradient dynamics (Eq. 13), stationary convergence analysis, and Lipschitz stability bounds demonstrate that MaPPO has strictly smaller gradient Lipschitz constants than DPO, leading to provably more stable training.

2. Practical Design with No Extra Hyperparameters:
Crucially, MaPPO introduces no new tunable hyperparameters. Reward gaps come directly from a reward model. This makes the method plug-and-play and broadly compatible with existing pipelines.

3. Compatibility with key DPO variants: compatible with both offline DPO (matching the classical setting) & Online I-DPO (by incorporating reward model scoring within each iteration). The authors also provide modified loss functions for SimPO, IPO, and CPO, demonstrating that the MaP term simply replaces the MLE component while leaving the rest intact.

**Audience:**

Yes

**Audience Explanation:**

Preference optimization (PO) is currently one of the most active research areas for LLM alignment. DPO and its variants (SimPO, ORPO, CPO, IPO) are highly influential, and MaPPO offers a theoretically principled extension that corrects well-known issues in DPO (confidence degeneration, near-tie over-penalization), improves stability and calibration, and integrates seamlessly with existing PO frameworks,

The gain will be acknowledged since it follows community standard.

**Broader Impact Concerns:**

No ethical concerns.

**Claims And Evidence:**

Yes

**Claims Explanation:**

The paper provides strong empirical evidence across 5 model families, 6 comprehensive benchmarks, 3 standard alignment tasks, and multiple DPO variants within diverse settings. Algorithm 1 and Figure 3 clearly illustrate the iterative tuning loop.

The improvements are statistically meaningful and consistent, especially on AlpacaEval 2.0 and Arena-Hard, where MaPPO often provides double-digit gains. The degradation patterns of DPO (squeezing effect) are demonstrated using concrete log-prob examples (Table 1: yw logprob reduced from −14.3 to −121.5).
The theoretical analysis (stationary convergence, Lipschitz stability, and gradient structure) is technically sound and well-motivated.

**Requested Changes:**

Very minor, suggested changes: Discuss/categorize most recent variants since submission, making it more tutorial for non-expert researchers. For instance: DAPO[1], GSPO[2], PRO[3].

[1] Yu, Qiying, et al. "Dapo: An open-source llm reinforcement learning system at scale." NeurIPS 2025.

[2] Zheng, Chujie, et al. "Group sequence policy optimization." arXiv preprint arXiv:2507.18071 (2025).

[3] "Proximalized Preference Optimization for Diverse Feedback Types: A Decomposed Perspective on DPO." NeurIPS 2025.

---

Checked that the updated version has included the requested changes.

---

> ### Author Response · Authors · 2025-12-10
> **Rebuttal by Authors**
>
> We sincerely appreciate the reviewer’s valuable and constructive feedback. In accordance with the suggestion, we have added a paragraph in Appendix D.2 that discusses and categorizes several recent RL-based methods for LLM post-training, including DAPO [1], GSPO [2], and PRO [3]. We thank the reviewer for the positive comments and for helping us improve the clarity and completeness of the manuscript.

---

> > ### Comment · Reviewer_2SVh · 2025-12-10
> > **Reply to Rebuttal**
> >
> > Thanks the authors for the revision. The updated version looks good to me.

---

### Comment · Action_Editor_k4MA · 2025-12-20
**Mistake in Section 4.2**

The topic of the manuscript is indeed quite interesting, and I also took a quick read of the paper. I do have one question regarding some of the claims made in the paper, Section 4.2 in particular.

When discussing the connection of MaPPO and SFT, the authors claimed that due to the monotonicity of $\log \sigma(\cdot)$, the optimal solutions of Eq.(9) and Eq. (10) will be the same. I don't think that's the case in general, since the $\log\sigma(\cdot)$ is applied **within** the expectation, which will change the per-sample loss differently due to the nonlinearity of $\log\sigma$. It is not hard to come up with a counterexample to refute the authors' claim:

> Consider there is one prompt $x$, with two responses $y_1$ and $y_2$. In the underlying comparison dataset half of the time $y_1$ is the winner and half of the time $y_2$ is the winner. Suppose for the given $\pi_{\mathrm{ref}}$ the distributions over $y_1$ and $y_2$ is skewed: $\pi_{\mathrm{ref}}(y_1 | x) = 0.9$ and $\pi_{\mathrm{ref}}(y_2 | x) = 0.1$. In this case the SFT solution will be uniform over $y_1$ and $y_2$, whereas the opt in Eq.(9) won't.

The authors need to fix the issue before the manuscript can be published.

---

> ### Author Response · Authors · 2025-12-23
>
> Thanks you for pointing out this mistake. We indeed overclaimed this connection in the original manuscript. We have modified Section 4.2. Analysis of MaPPO in color blue.
>
> In short, the stochastic gradient of MaPPO ($\Delta_{r} = 0$) has the same direction as the stochastic gradient of SFT, instead of the optimal points.
>
> The dominator and $\beta$ is from the KL divergence constraint and regulates the model $\pi_{\theta}$ to behave close to the initial model $\pi_{\rm ref}$, which is $\pi^{\star}(y_{1}|x) > \pi^{\star}(y_{2}|x)$. If $\beta$ is close to $0$, $\pi^{\star}(y_{1}|x)$ and $\pi^{\star}(y_{2}|x)$ are close to $0.5$, and if $\beta$ is very large, $\pi^{\star}(y_{1}|x)$ and $\pi^{\star}(y_{2}|x)$ are close to $0.9$ and $0.1$.

---

> > ### Comment · Action_Editor_k4MA · 2025-12-23
> > **Still misleading and not accurate**
> >
> > Unfortunately, I am still not convinced by the revised claims, as they're misleading.
> >
> > By claiming "the stochastic gradient of MaPPO has the same direction as the stochastic gradient of SFT", if my understanding is correct, the authors intended to mean the case where batch size = 1, i.e., for a single triplet $(y_w, y_l, x)$, the gradient of MaPPO loss and the SFT loss over the model parameters for this triplet are proportional to each other. If that's the authors' intention, then please just state so explicitly by providing the derivation for a single triplet under the two losses, and remove the expectations from Eq. (9) to (11), since under expectation (even for the case of mini-batch with batch size > 1), their SGD directions will in general be different. Given the weak connection, I'd suggest that the authors completely remove this paragraph.
> >
> > Another clarification point: even in the online RL case, this is not the rejection sampling method in Dong et al. 2023. This claim needs to be removed as well.

---

### Decision · Action_Editor_k4MA · 2025-12-23

**Recommendation:** Reject

**Audience:**

Yes

**Audience Explanation:**

The overall topic of the paper is indeed quite interesting and relevant to reward modeling in RLHF.

**Claims And Evidence:**

No

**Claims Explanation:**

There is one major claim on the connection between the proposed MaPPO method and SFT made by the authors that is not correct in general. The AE has pointed out the mistake during the decision-making stage. The AE discussed the found mistake with reviewers and authors, and both confirmed the mistake.

The authors then responded with revised claims, but the revised claims are still not accurate and are potentially misleading. Given the discussion required to address this major claim, I will reject the manuscript for now and encourage the authors to thoroughly address the issues and then resubmit.

**Resubmission Of Major Revision:**

The authors may consider submitting a major revision at a later time.